# Lateral Border of a Small River Plume: Salinity Structure, Instabilities and Mass Transport

Alexander Osadchiev [1,2,*], Alexandra Gordey [1,2], Alexandra Barymova [3], Roman Sedakov [1], Vladimir Rogozhin [1,3], Roman Zhiba [4] and Roman Dbar [4]

1 Shirshov Institute of Oceanology, Russian Academy of Sciences, Nakhimovskiy Prospect 36, 117997 Moscow, Russia
2 Moscow Institute of Physics and Technology, Institusky Lane 9, 141701 Dolgoprudny, Russia
3 Marine Research Center at Lomonosov Moscow State University, Leninskie Gory 1, 119992 Moscow, Russia
4 Institute of Ecology, Academy of Sciences of Abkhazia, Krasnomayatskaya Str. 67, 384900 Sukhum, Abkhazia
* Correspondence: osadchiev@ocean.ru

**Abstract:** The interfaces between small river plumes and ambient seawater have extremely sharp horizontal and vertical salinity gradients, often accompanied by velocity shear. It results in formation of instabilities at the lateral borders of small plumes. In this study, we use high-resolution aerial remote sensing supported by in situ measurements to study these instabilities. We describe their spatial and temporal characteristics and then reconstruct their relation to density gradient and velocity shear. We report that Rayleigh–Taylor instabilities, with spatial scales ~5–50 m, are common features of the sharp plume-sea interfaces and their sizes are proportional to the Atwood number determined by the cross-shore density gradient. Kelvin–Helmholtz instabilities have a smaller size (~3–7 m) and are formed at the plume border in case of velocity shear >20–30 cm/s. Both instabilities induce mass transport across the plume-sea interfaces, which modifies salinity structure of the plume borders and induces lateral mixing of small river plumes. In addition, aerial observations revealed wind-driven Stokes transport across the sharp plume-sea interface, which occurs in the shallow (~2–3 cm) surface layer. This process limitedly affects salinity structure and mixing at the plume border, however, it could be an important issue for the spread of river-borne floating particles in the ocean.

**Keywords:** small river plume; plume border; plume-sea interface; Rayleigh–Taylor instability; Kelvin–Helmholtz instability; mixing

## 1. Introduction

River plumes are common features in many coastal regions of the World Ocean. River plumes are formed by lateral inflow of freshwater to the sea and are spreading in the sea surface layer, i.e., they remain at the sea boundary. As a result, river plumes mix slowly with ambient seawater and generate sharp salinity gradients at the plume-sea interface. These gradients are especially strong at plumes formed by small rivers, which have small spatial scales and small freshwater residence time equal to hours and days [1–3]. In particular, horizontal (3–4 salinity units/m [2,4]) and vertical (5–6 salinity units/m [5,6]) salinity gradients observed at the outer borders of small plumes are among the largest reported in the World Ocean.

The interactions between river plumes and ambient sea are important for understanding land-ocean fluxes of fluvial water and river-borne dissolved and suspended matter. Structure and circulation at the plume border play an important role in transport of fine terrigenous sediments [7–13] and floating matter, including river-borne marine litter and microplastic [14–17], fish larvae [18–21], etc. However, many processes at the plume-sea interface remain understudied, especially those with small spatial and temporal scales. In particular, only a few works have addressed the small-scale vertical structure and circulation at the plume border. Even the shape of the plume bottom at this area (as well as

elsewhere within a river plume) has never been measured or observed with good spatial resolution. The main constraining factor for this research consists of the complexity of in situ measurements at the shallow and extremely variable plume-sea interface.

Among the main findings about the processes at the plume border are: the downwelling circulation and the resulting surface convergence at the lateral plume border [22], mixing estimates caused by velocity shear and tidal forcing [23–28], and formation of coherent flow structures associated with different types of instability (Kelvin–Helmholtz, Rayleigh–Taylor, Holmboe) [29–36]. The instabilities at the lateral plume border were first reported in 1974 [37], however, their observations and measurements at real plumes are still very scarce (only five river plumes, namely, Connecticut, Merrimack, Elwa, Kodor, and Bzyb plumes). Direct measurements of thermohaline structure and circulation at the plume border are also very scarce [2,26].

Coherent flow structures in gravity currents were addressed in many studies during recent decades [38]. Nevertheless, there is a large gap between theoretical solutions, numerical and laboratory modeling of frontal instabilities at stratified fluids, on the one hand, and observations and measurements of frontal instabilities in river plumes, on the other hand [39,40]. The results of the majority of these works could be limitedly applied to plume-sea interfaces and we are not aware of any theoretical model or numerical parameterization that could predict sizes and dynamics (formation, evolving and merging) of instabilities at the plume border.

In situ measurements represent ground truth for studies of processes in river plumes. On the other hand, spatial and/or temporal resolution of discrete in situ measurements is relatively low. As a result, studies based only on in situ measurements often have the inherent spatial or temporal limitations. Satellite and aerial remote sensing could support in situ measurements and substantially increase spatial coverage of the considered processes including frontal instabilities. However, remote sensing observations are limited to surface manifestations of these processes and do not resolve the vertical structure of river plumes.

On the contrary, theoretical solutions, numerical and laboratory modeling reproduces the three-dimensional plume structure with relatively high spatial and temporal resolution. The main limitation of these studies consists of the fact that they require thorough validation against in situ data (which is often lacking) in order to verify that they represent processes which occur in real river plumes. Therefore, precise field observations are still necessary to determine the physical background of instabilities at the plume-sea interface and support the related numerical modeling.

In situ salinity and velocity measurements by CTD and ADCP profilers give the main information about the structure and dynamics of the plume-sea interface. However, recent development of quadcopters provided fundamentally new opportunities to observe and measure processes at the lateral plume border, which are visible by aerial remote sensing [2,41]. Usage of autonomous underwater vehicles [42,43] and drifters [27] also showed a certain progress in measurements at the bottom plume border.

The main straightforward characteristics, which are used to study instabilities in river plumes, include, first, the spatial and temporal characteristics of these instabilities (wavelength, amplitude, motion speed, vorticity, residual time, etc.) and, second, the characteristics of gradients at the plume-sea interface, which govern formation of instabilities (density gradient and velocity shear) [2,29,30,32,33,41]. A large set of characteristics, which describe the role of instabilities in turbulent mixing (eddy viscosity, vertical diffusivity, eddy kinetic energy, buoyancy dissipation, etc.), is mainly addressed and analyzed in numerical modeling studies due to the complexity of in situ turbulence measurements in river plumes [30,34,36,44,45]. Finally, a number of dimensionless numbers (Rossby, Richardson, Reynolds, Atwood, Grashof, Ertel, etc.) are widely used to compare mass, momentum, and energy transport associated with instabilities in different laboratory and natural systems [46–53].

This study describes observations and measurements during five field surveys at two small river plumes located in the Black Sea. These field surveys were performed

during weak external forcing conditions, i.e., almost no tide, light wind, small waves, and light ambient coastal circulation. It provided the opportunity to assess, through field observations (as opposed to numerical modeling), the effects of any particular forcing when it is at maximum strength. In this study, we focus on small-scale thermohaline structure and dynamical processes at the plume-sea interface related to formation of Rayleigh–Taylor (due to density gradient) and Kelvin–Helmholtz (due to velocity shear) instabilities.

This study is organized as follows. In Section 2, we describe the study area, the in situ measurements and aerial observations, and the methods of processing of remote sensing data. In Section 3, we describe the salinity structure at the borders of the considered river plumes and reconstruct spatial and temporal characteristics of frontal instabilities. In Section 4, we discuss the physical background of the observed features at the plume border and assess its influence on transport of dissolved (salt) and suspended (terrigenous sediments and floating litter) matter across sharp plume-sea interface, which is followed by the conclusions in Section 5.

## 2. Data and Methods

### 2.1. Aerial Observations and In Situ Measurements

Five field surveys described in this study were performed at the buoyant plumes formed by the Kodor and Bzyb rivers in the eastern part of the Black Sea (Figure 1). General characteristics of the Kodor and Bzyb plumes and the ambient saline sea are described in detail by Osadchiev et al. [2–4,6]. The information about all field surveys analyzed in the study is summarized in Table 1.

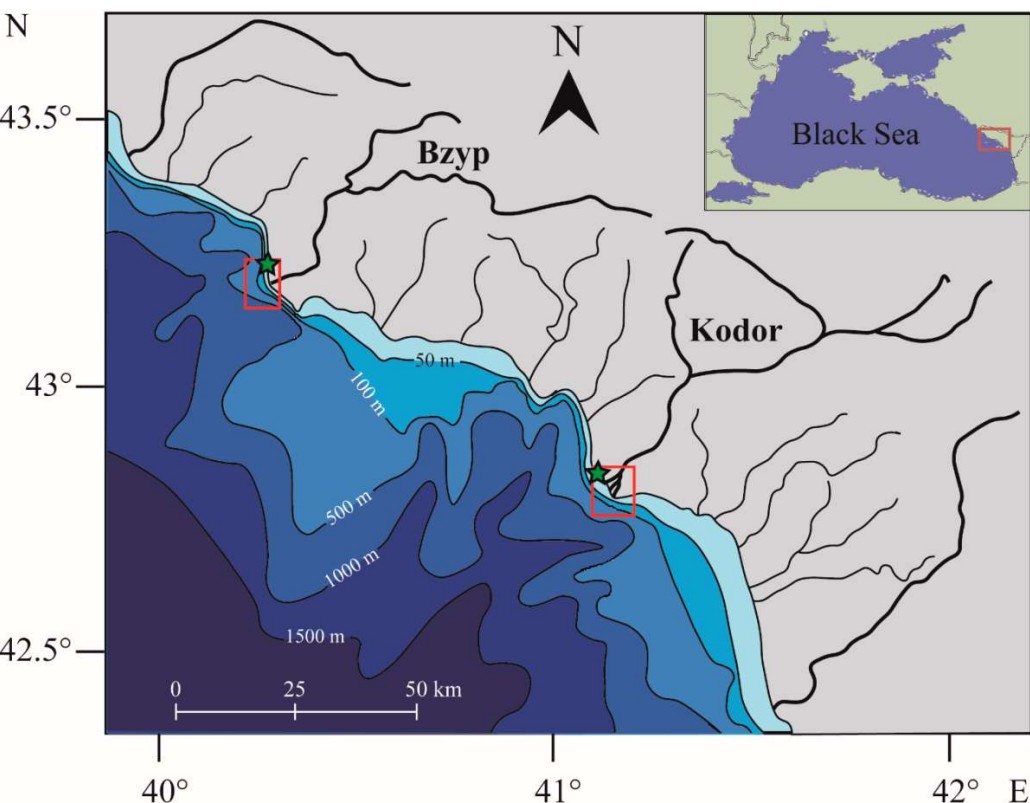

**Figure 1.** Study areas at the Bzyb and Kodor plumes at the eastern part of the Black Sea indicated by red boxes. Green stars indicate locations of meteorological stations.

**Table 1.** Periods, areas of field work, and types of in situ measurements of oceanographic surveys.

| Period | Area of Field Work | In Situ Measurements |
|---|---|---|
| 1–2 September 2018 | Kodor plume | CTD (*SBE 911plus*) and ADCP (*Teledyne RDI Workhorse Sentinel*) |
| 1–4 April 2019 | Kodor plume | CTD (*SBE 911plus*) and ADCP (*Teledyne RDI Workhorse Sentinel*) |
| 31 May–1 June 2019 | Bzyb plume | CTD (*SBE 911plus*) |
| 14–18 April 2021 | Bzyb plume | CTD (*YSI CastAway*) and ADCP (*Nortek AquaDopp*) |
| 27–30 April 2022 | Bzyb plume | CTD (*YSI CastAway*) |

The general workflow of our field surveys is the following. First, we performed aerial remote sensing of the plume using quadcopters. Aerial observations provided the initial information about the plume shape, location of its borders, and presence of border instabilities. Then we collected aerial imagery and/or video records of detected instabilities. Second, synchronously to remote sensing, we performed in situ measurements of thermohaline and velocity structure across the plume border. CTD measurement were organized either as surface to bottom profiling along the cross-border transect with high spatial resolution or continuous measurements at fixed depths during drift of a boat across the plume border. ADCP measurements were performed either from a stable bottom-mounted profiler or moving boat-mounted profiler with bottom tracking. Third, we scattered floating natural tracers (sawdust and corncobs) from a boat at different parts of the plume-sea interface and were tracing their motion by aerial remote sensing. The resulting data set collected during field surveys provided the necessary information about thermohaline and velocity structure across the plume border, as well as spatial scales, circulation patterns, and velocities of plume border instabilities.

Aerial observations of the Kodor and Bzyb plumes were performed by *DJI Mavik 2 Zoom* and *DJI Phantom 4 Pro* quadcopters equipped with a 12 MP/4K video cameras. Quadcopter shooting altitude depended on the spatial scale of the sensing sea surface process and varied from 10–30 m for the small-scale frontal circulation to 150–200 m for detection of plume position and area. Weather conditions during the field surveys were favorable for usage of the quadcopter. The flights were conducted during no-rain conditions from morning to evening. In cases with clear sky conditions, sun glint strongly affected the quality of the aerial data during the daytime. The remote sensing data was georeferenced by direct projection into an earth-based Cartesian coordinate system using GPS and altimetry data from the quadcopter. The accuracy of the direct projection was validated and corrected according to the known sizes of the R/V registered in the imagery.

Thermohaline measurements within the Kodor and Bzyb plumes were performed by the *SBE 911plus* and *YSI CastAway* CTD instruments, while ADCP measurements were performed by the *Teledyne RDI Workhorse Sentinel* and *Nortek AquaDopp* ADCP profilers (Table 1). The aerial observations and in situ measurements were supported by synchronous wind measurements by a *Gill GMX200* sensor, which was located at an altitude of 8–10 m on the seashore near river mouths.

The discharge rates of the Kodor and Bzyb rivers were reconstructed using the indirect method based on satellite observations and Lagrangian numerical modeling of river plumes. The general idea of this method is reconstruction of configuration of external forcing conditions (including river discharge rate) for a numerical model, which provides a river plume similar to that observed at satellite imagery. This method was validated against gauge measurements at small rivers in the study area, which was described in detail in [54]. The method was additionally validated for the Kodor plume by in situ discharge measurements performed during a hydrological field survey in August–September 2018.

*2.2. Processing of Aerial Data*

Aerial images were processed using the open-source computer vision library *OpenCV* (https://opencv.org, accessed on 1 June 2022). All images were converted to grayscale, then the brightness of pixels was analyzed. The border between saline seawater and river plumes was detected by the Canny edge method based on the search for maximum brightness gradients. This algorithm was chosen due to two main reasons: it has low computational cost and it demonstrated accurate detection for our dataset. The dataset consisted of pictures, which generally contain only two "objects": turbid (brown) plume and clean (blue) seawater. These "objects" have a large difference in color, so the typical picture was a two-colored image with clear simple-shaped border. As the brightness gradient between these two water masses was high, the border was efficiently detected by the candy edge algorithm. What is more important, the Canny edge detector was relatively insensible to local minima, which was frequently registered at the plume-sea border due to increased turbulent mixing and presence of floating matter and foam.

The sizes of eddies were estimated using the following procedure. Several images of typical clefts in different spatial scales were chosen to be used as templates of a cleft. Every template was superimposed over the processed picture and the similarity R between the template and the current subarea of a whole picture was calculated in the following way. $T(x, y)$ denotes the template brightness in point $(x, y)$, $I(x, y)$—the brightness of the image. $T'(x, y)$, $I'(x, y)$ are deviations of the brightness from its mean value for the template and image, respectively:

$$T'(x', y') = T(x', y') - \frac{\sum_{x^*, y^*} T(x^*, y^*)}{wh},$$

$$I'(x + x', y + y') = I(x + x', y + y') - \frac{\sum_{x^*, y^*} I(x + x^*, y + y^*)}{wh},$$

where $(x', y')$ is the current point in the template; $(x, y)$ is the point in the image, where the template is applied to it; $w, h$ are the width and the height of the template. Then the similarity $R$ at every point of the picture is

$$R(x, y) = \frac{\sum_{x', y'} T'(x', y') \cdot I'(x + x', y + y')}{\left(\sum_{x', y'} T'(x', y')^2 \cdot I'(x + x', y + y')^2\right)^{\frac{1}{2}}}.$$

Since subareas with the highest similarity value correspond to clefts, which separate lobes, the distance between the neighboring detected clefts was considered as the size of the lobe between these clefts.

Particles trajectories and their velocities were obtained with the Gunnar–Farneback optical flow method, the main idea of which is that the brightness of an object remains almost constant during its motion. This assumption enables us to track an object in serial images by searching for the similar brightness pattern in the area, where it was detected in the previous image. This method provides object displacements, from which particles velocities were calculated.

## 3. Results

The field surveys at the Kodor and Bzyb plumes were performed during different discharge conditions, namely, low discharge on 1–4 April 2019 (Kodor River before spring freshet, 40 m$^3$/s); medium discharge on 2 September 2018 (Kodor River one day after rain-induced flash flood, 80 m$^3$/s) and on 14–18 April 2021 (Bzyb River during the beginning of spring freshet, 100 m$^3$/s); high discharge on 1 September 2018 (Kodor River several hours after a rain-induced flash flood, 150 m$^3$/s), on 31 May–1 June 2019 and 27–30 April 2022 (Bzyb River during well-developed spring freshet, 180–200 m$^3$/s). Wind forcing was moderate during these field surveys. Average and maximal wind speed registered at weather stations in the study regions were 2–3 and 5–8 m/s during field surveys. Tidal circulation at the study area is very low and tidal amplitudes are less than 6 cm [55,56].

Coherent lobe-and-cleft structures at the sharp plume-sea interface previously described for the Kodor and Bzyb plumes by Osadchiev et al. [2] were observed by aerial remote sensing during all field surveys. However, these structures had different typical sizes during different field surveys (Figure 2).

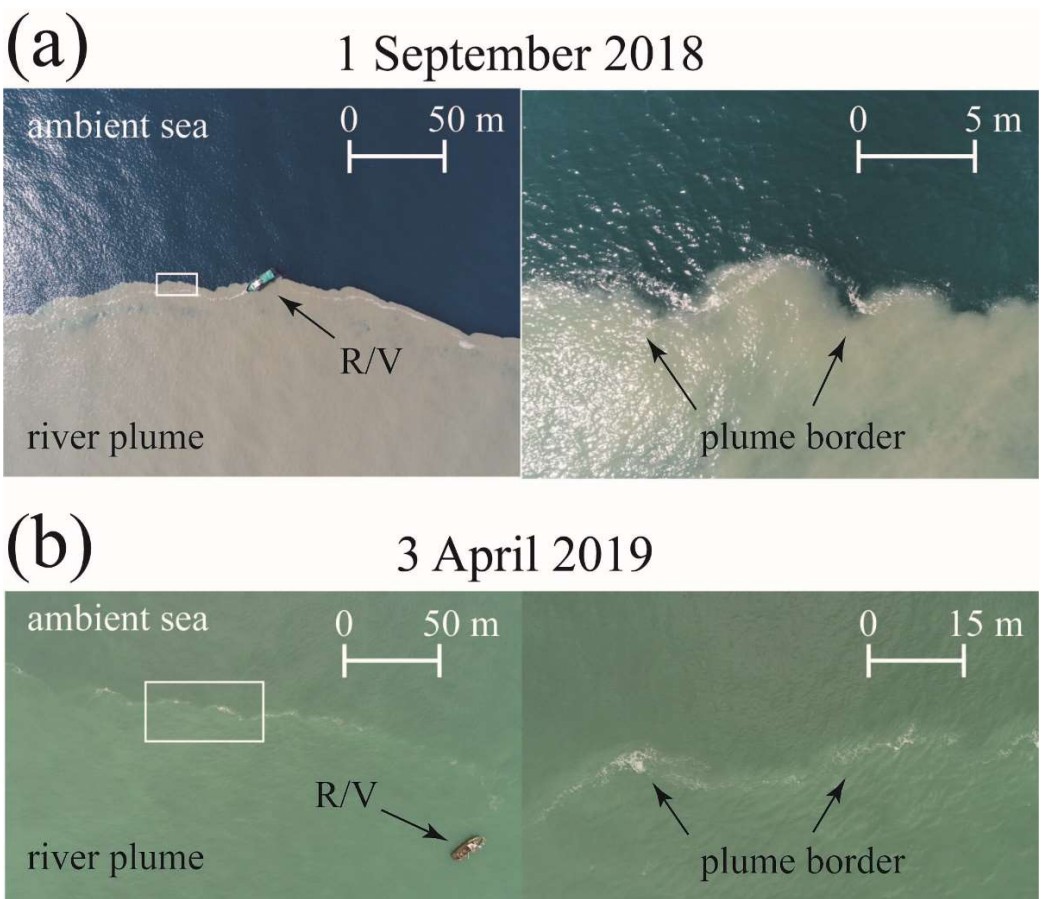

**Figure 2.** Aerial images of the Kodor plume border on: (**a**) 1 September 2018 and (**b**) 3 April 2019 illustrating different spatial scales of frontal instabilities. White boxes in the left panels indicate zoomed areas shown in right panels.

The smallest lobes (3–10 m) were registered in the near-field and far-field parts of the plume during high-discharge events, e.g., on 1 September 2018, 1 July 2019, 15 April 2021, 27 April 2022 (Figure 2a). Large lobes (30–60 m), on the contrary, were observed in the far-field parts of the plume during different conditions including flooding periods (e.g., 30 April 2022), drought periods (e.g., 3 April 2019), and the period shortly after the rain-induced flash flood (e.g., 2 September 2018) (Figure 2a). Thus, sizes of lobe-and-cleft structures vary in time within individual river plumes and could be very different even during consecutive days if the external conditions changed significantly, e.g., during and after short-term flash floods on 1–2 September 2018.

Moreover, sizes of lobe-and-cleft structures are different at different sizes of an individual river plume. Continuous aerial recording along the Kodor and Bzyb plume borders revealed that spatial scales of these structures are the smallest near the river mouths and steadily increase along the border with an increase in the distance from the rivers (Figure 3). In particular, typical sizes of lobes along the Bzyb plume border on 28 April 2022 increased from 5–7 m on a distance of 50 m from the river mouth (Figure 3a) to 10–15 m on a distance of 150 m (Figure 3b) and then to 20–40 m on a distance of 400 m (Figure 3c).

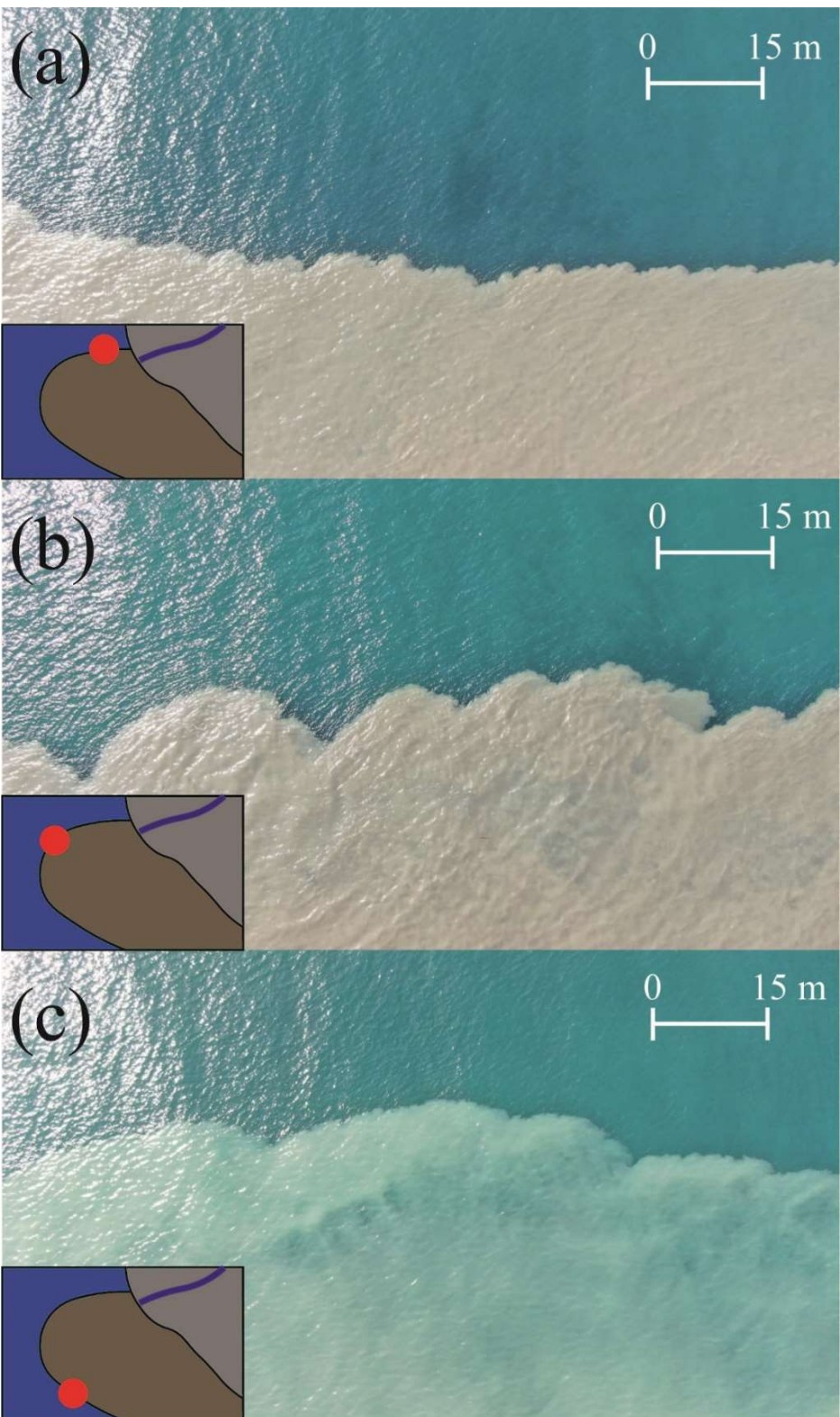

**Figure 3.** Aerial images of different parts of the Bzyb plume border on 28 April 2022 on distances of (**a**) 50 m, (**b**) 150 m, and (**c**) 400 m along the border from the river mouth illustrating different spatial scales of frontal instabilities. Red circles at the insets indicate segments of the plume border visible at the respective aerial images.

In order to study the physical background of the formation of lobe-and-cleft structure of the plume borders, aerial observations were accompanied by synchronous in situ measurements of thermohaline properties using CTD-instruments and current velocity using ADCP-profilers and the tracing of floating particles with aerial imagery and video records. As a result, the information about spatial scales of instabilities was supported by information about density gradients across the plume-sea interfaces and velocity shear between plume and ambient saline sea.

The vertical salinity and velocity structures across the near-field and far-field parts of the plume border are shown in Figures 4–6. The vertical and horizontal salinity structure at the plume-sea interface is strongly inhomogenous. The vertical salinity gradient at the bottom plume border was equal to 5–6 salinity units/m and relaxed to 2–3 salinity units/m in the vicinity of the lateral plume border (Figure 4b). The horizontal salinity gradient at the lateral plume border was smaller (~1 salinity units/m at the surface layer) and decreased with the depth (~0.2 salinity units/m at the depth of 1 m) (Figure 6b). Note that the structure of the sharp plume-sea interface with relatively small vertical (<5 m) and horizontal (<50 m) scales remained stable along large segments of the lateral plume border. The velocity gradient at the lateral plume border was large in the near-field part of the plume ($2$–$4 \times 10^{-3}$ 1/s) due to large inertia of the inflowing river jet as compared to the less energetic far-field part of the plume (~$1 \times 10^{-3}$ 1/s) (Figure 6). Note that the measurements were performed mostly during the low wind forcing periods, therefore ambient coastal currents were also very small (~0.05 m/s).

Aerial imagery regularly detected stripes of low-turbid water within the far-field parts of the Kodor and Bzyb plumes, which were stretched along the lobe-and-cleft plume borders (Figure 5a). Osadchiev et al. [2] presumed that these low-turbid stripes are formed as a result of merging instabilities at the plume borders and the resulting cross-border salt transport. In order to study the fine-scale salinity structure at the sharp plume-sea interface and to prove this assumption, we performed multiple high-resolution CTD measurements across the sharp plume borders of the Kodor and Bzyb plumes. We revealed that these low-turbid stripes indeed have increased salinity by 2–5 units as compared to both the area between the stripe and the plume border and area shoreward from the stripe towards the river mouth. In particular, field survey at the Bzyb plume border on 15 April 2021 showed that this low-turbid stripe was located 20 m from the plume border and was 5–8 m wide (Figure 5a). Salinity within the stripe (8–9) as compared to surrounding plume (6–7) (Figure 5b). However, the depth of the vertical plume-sea interface within this stripe (1–1.5 m) was the same as in the surrounding plume. Simpson et al. [41] also reported even greater increase in salinity in the plume in the vicinity of the lobe-and-cleft plume border from 5 to 13 (black line in Figure 8b in [41]).

Circulation at the plume-sea interface and velocity shear across the plume border was studied using direct ADCP velocity measurements and tracking of floating matter at the plume border. During the field surveys, we performed experiments by scattering natural tracers at different parts of the plume-sea interface and tracing their motion by aerial remote sensing. For this purpose, we used thin sawdust, which represented suspended matter (e.g., terrigenous sediments), and 10–20 cm long corncobs, which represented large floating particles (e.g., plastic pollution). Sawdust and corncobs were scattered along and across the plume border to study, first, circulation at the lobe-and-cleft instabilities and, second, mass transport across the plume-sea interface and convergence of matter at the plume border.

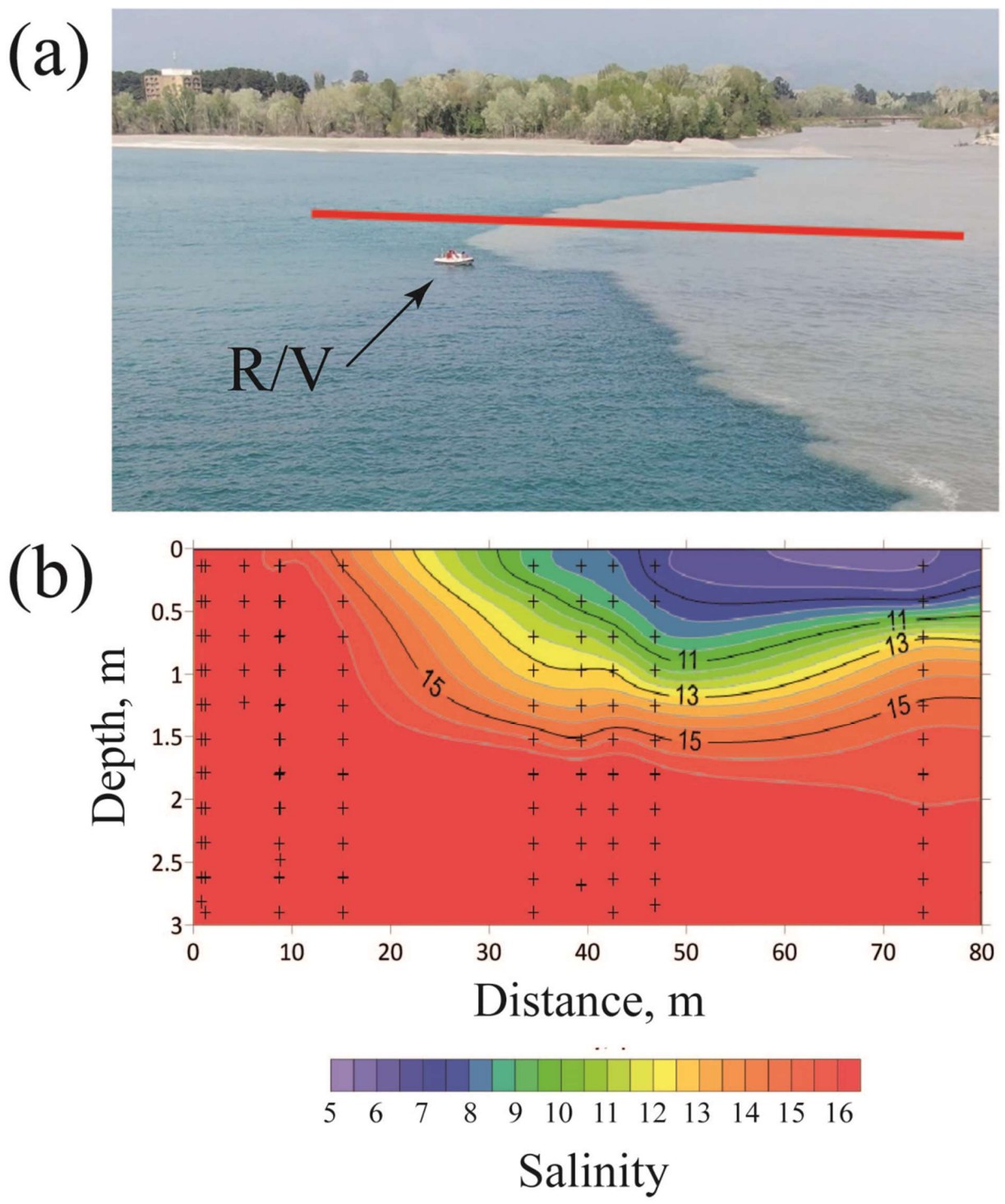

**Figure 4.** Sharp near-field plume border visible at: (**a**) aerial image of the Bzyb plume border and (**b**) salinity structure across the Bzyb plume border on 27 April 2022. Red line in panel (**a**) indicates location of the cross-border transect. Black crosses in panel (**b**) indicate points of in situ measurements.

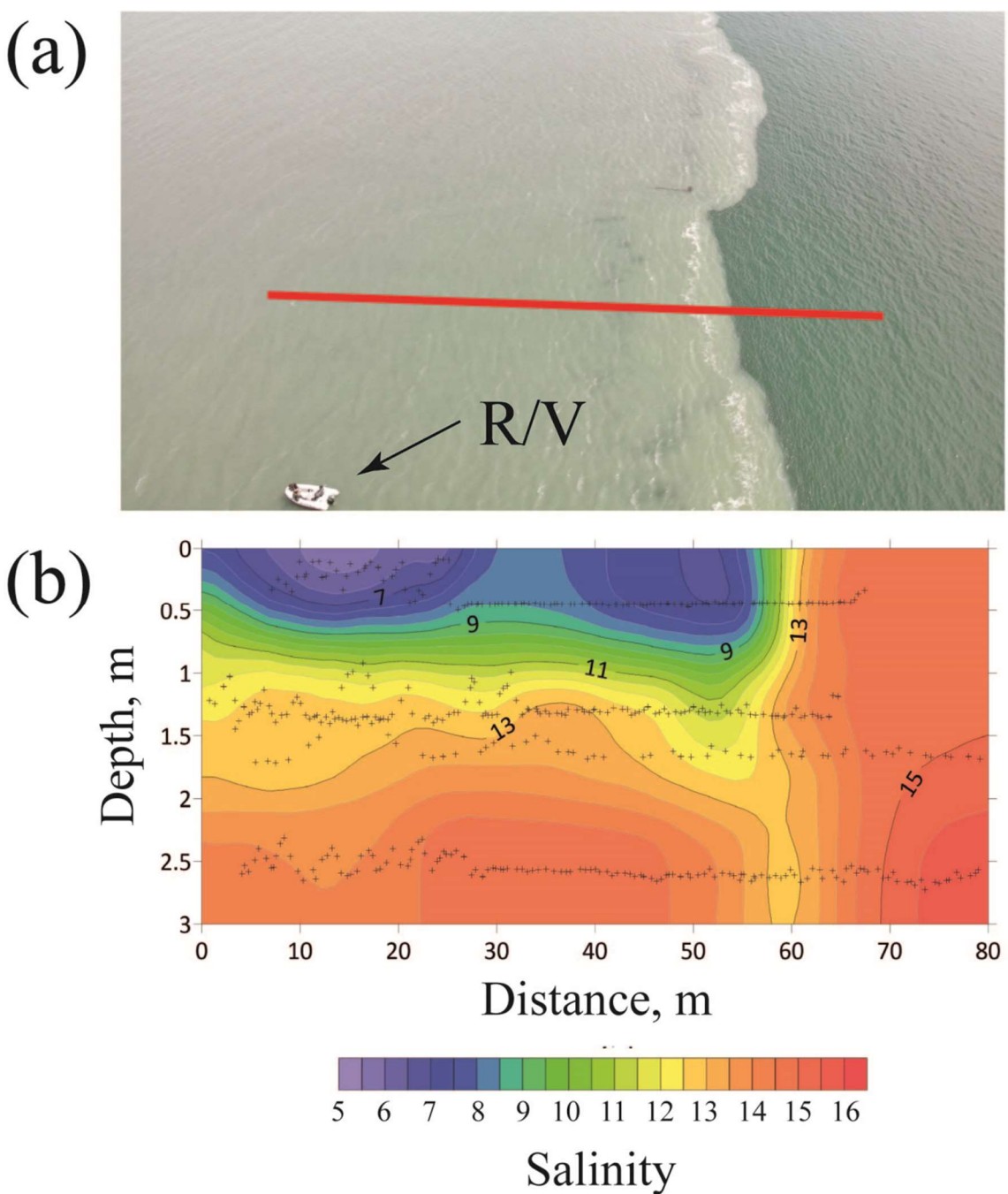

**Figure 5.** Stripe of reduced salinity and turbidity formed along the far-filed part of the plume border visible at: (**a**) aerial image of the Bzyb plume border and (**b**) salinity structure across the Bzyb plume border on 15 April 2021. Red line in panel (**a**) indicates location of the cross-border transect. Black crosses in panel (**b**) indicate points of in situ measurements.

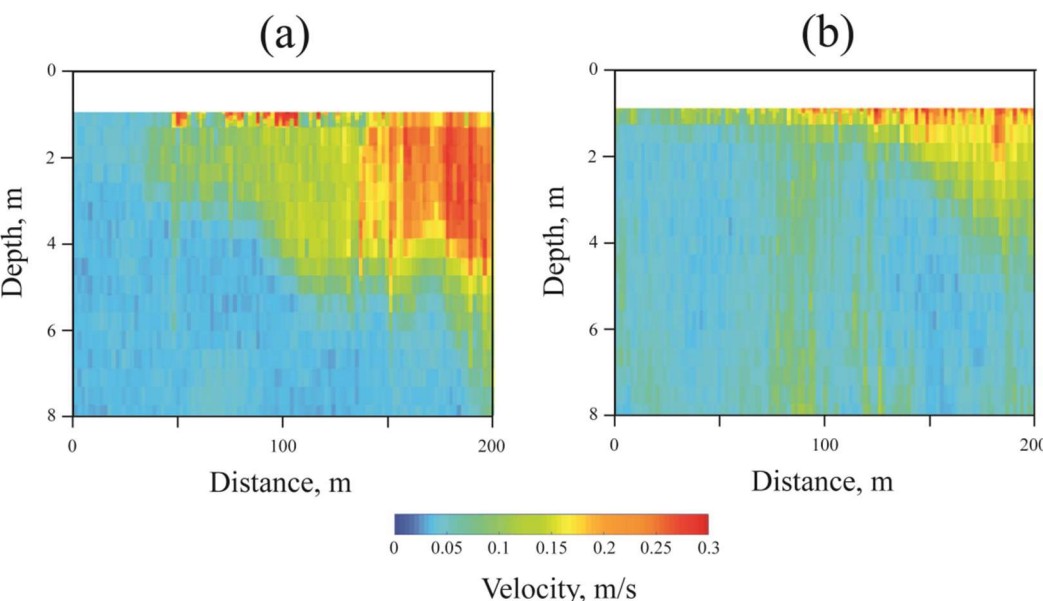

**Figure 6.** Absolute velocity structure across the near-field (**a**) and far-field (**b**) parts of the Kodor plume border on 3 April 2019.

The "along-border" experiments were performed in the near-field (Figure 7) and far-field (Figure 8) parts of the Bzyb plume and showed significantly different circulation patterns at the lobe-and-cleft instabilities. In both cases, initially tracers were regularly distributed along the plume border (Figures 7a and 8a). At the near-field part of the plume with large velocity shear (>20–30 cm/s) between river plume and ambient sea, all tracers were transported off the river mouth along the plume border (Figure 7b,c). As a result, their regular distribution along the border remained stable, e.g., distribution was similar the beginning of the "along-border" experiment (Figure 8a), after 9 min (Figure 8b), and after 17 min (Figure 8c). The offshore advection of tracers was accompanied by their anticyclonic circulation within small lobes (~3–7 m) at the near-field part of the plume border.

At the far-field part of the plume with small velocity shear (<10 cm/s), the tracers were advected towards the neighboring clefts in both directions along the plume border (Figure 8b). As a result, during 4–5 min from the beginning of the "along-border" experiment all tracers were accumulated within several patches (Figure 8c) and their initial regular distribution was completely distorted. Then the tracers remained trapped within clefts circulating in opposite directions, i.e., cyclonic/anticyclonic in left/right sides of the lobes.

The "cross-border" experiments were performed only in the far-field part of the Bzyb plume (Figure 9). Initially, tracers were regularly distributed along the transect, which started in the ambient sea near the plume border and ended within the plume 30 m far from its border (Figure 9a). Tracers from both sides of the plume border were advected directly towards the border (Figure 9b) illustrating convergence at this area. As a result, 3 min after the beginning of the "cross-border" experiment, all tracers were accumulated at the plume border (Figure 9c).

One of the "cross-border" experiments was performed with sawdust, which was scattered across the Bzyb plume border within the plume and ambient sea (Figure 10). The main feature of this experiment was the following. It was performed under moderate wind (8 m/s) blowing from the ambient sea towards the plume across the border. These wind forcing conditions resulted in an unexpected transport pattern of sawdust. Sawdust scattered at the ambient sea was advected towards the plume border (Figure 10a,b). Then the sawdust contained in the thin (several centimeters) surface layer crossed the plume border and remained at the surface, while the rest of the sawdust (contained in deeper layer) was downwelled below the plume (Figure 10c,d). Then the surface sawdust was

stretched into lines and advected further within the plume off its border (Figure 10e,f). We associate this cross-border transport of sawdust with wind-induced Stokes drift, which affected thin sea surface layer. Note, that during all other experiments with low wind forcing the tracers (sawdust and corncobs) remained accumulated at the plume border and did not cross it.

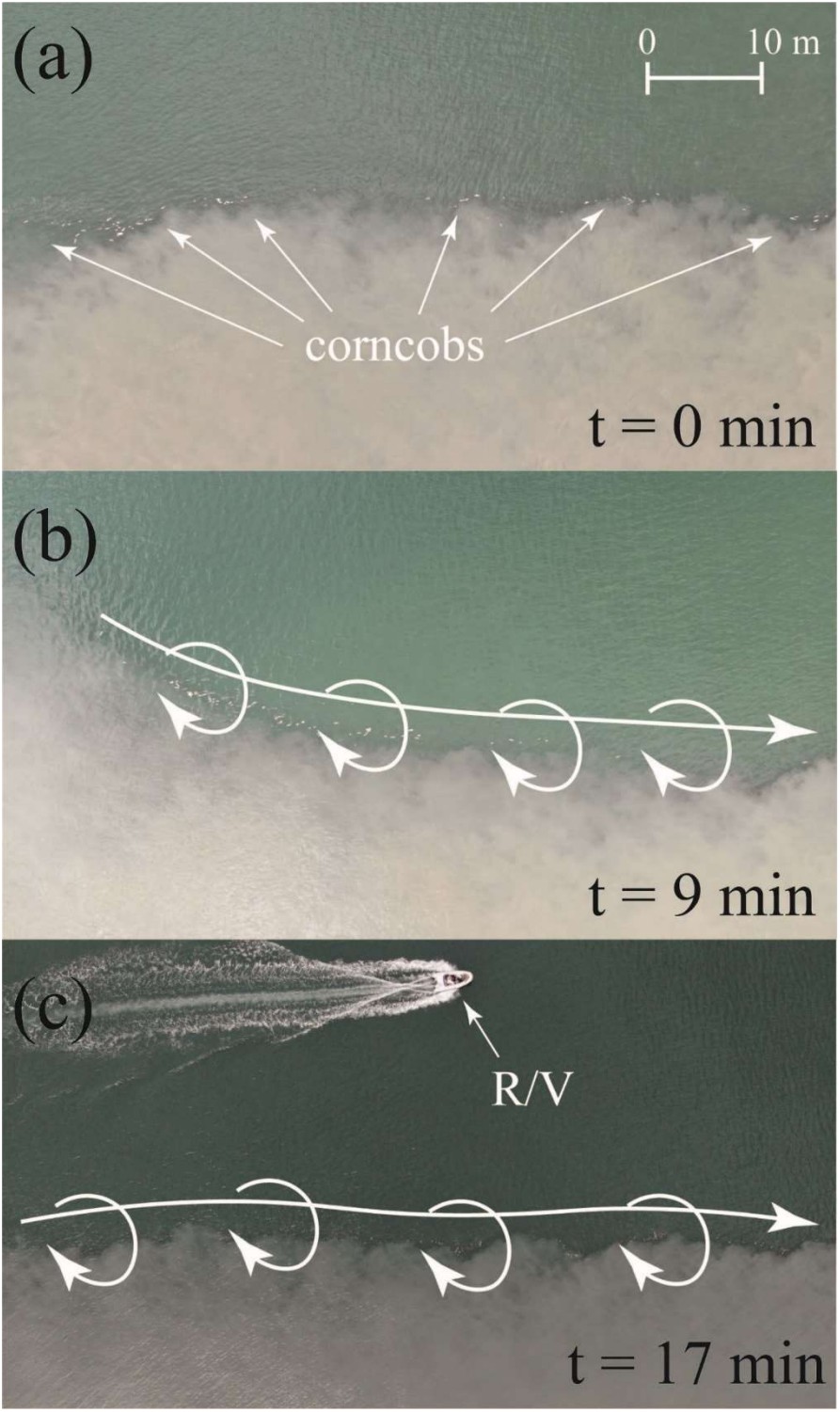

**Figure 7.** Aerial image of the near-field part of the Bzyb plume border illustrating motion of corncobs on 18 April 2021 at (**a**) the beginning of the "along-border" experiment, (**b**) after 9 min, and (**c**) after 17 min. White arrows in panels (**b**,**c**) indicate motion of corncobs.

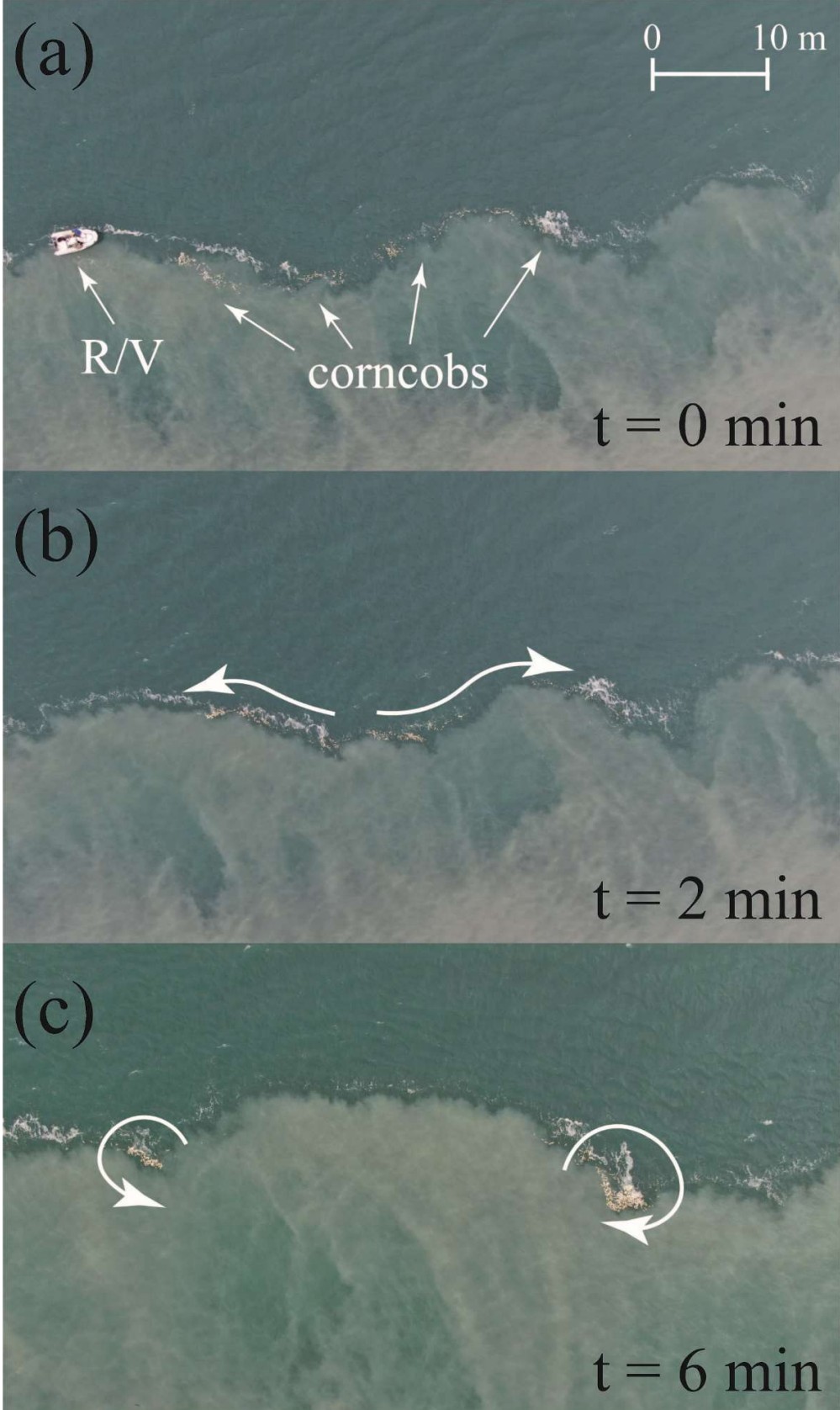

**Figure 8.** Aerial image of the far-field part of the Bzyb plume border illustrating motion of corncobs on 30 April 2022 at (**a**) the beginning of the "along-border" experiment, (**b**) after 2 min, and (**c**) after 6 min. White arrows in panels (**b**,**c**) indicate motion of corncobs.

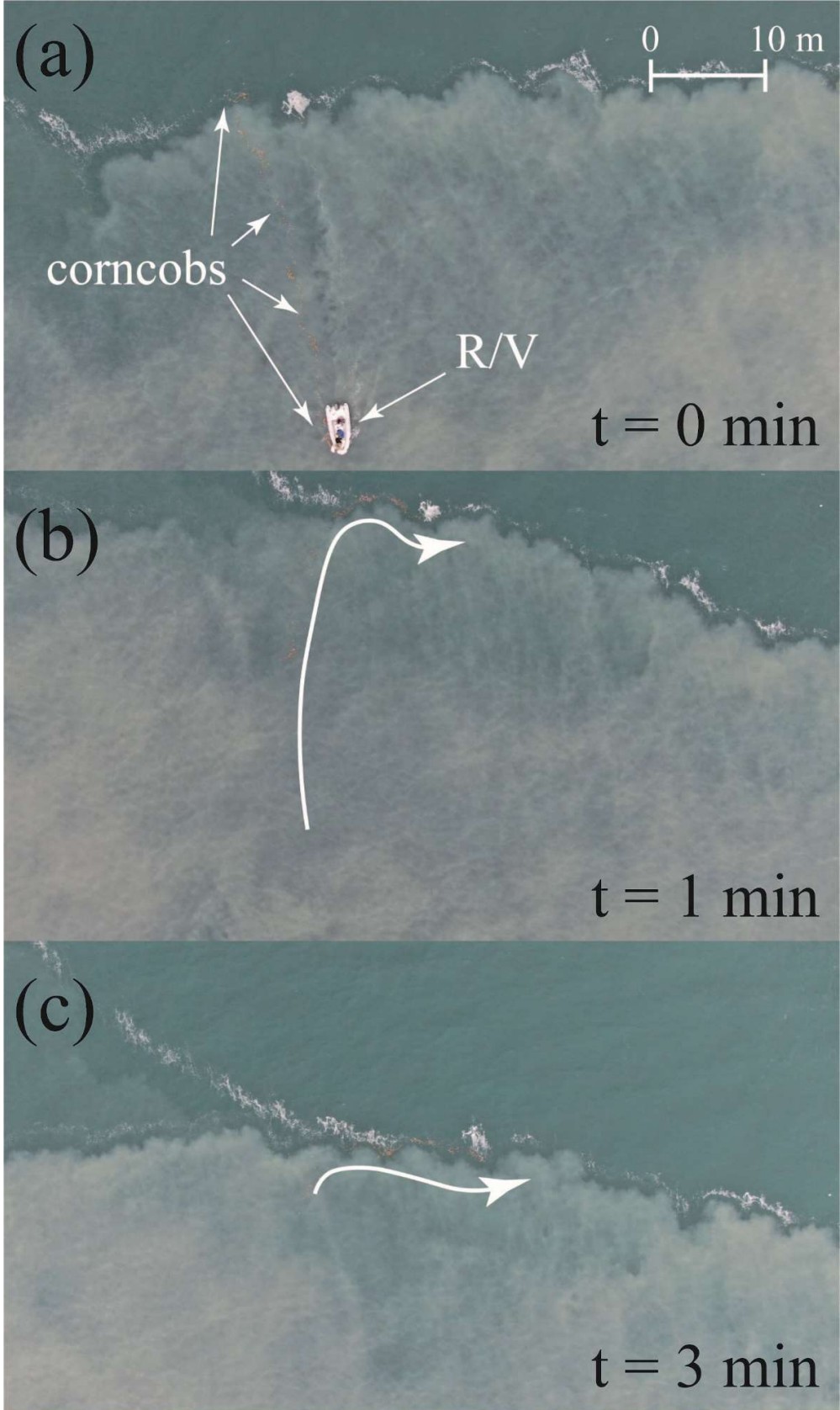

**Figure 9.** Aerial image of the far-field part of the Bzyb plume border illustrating motion of corncobs on 30 April 2022 at (**a**) the beginning of the "cross-border" experiment, (**b**) after 1 min, and (**c**) after 3 min. White arrows in panels (**b**,**c**) indicate motion of corncobs.

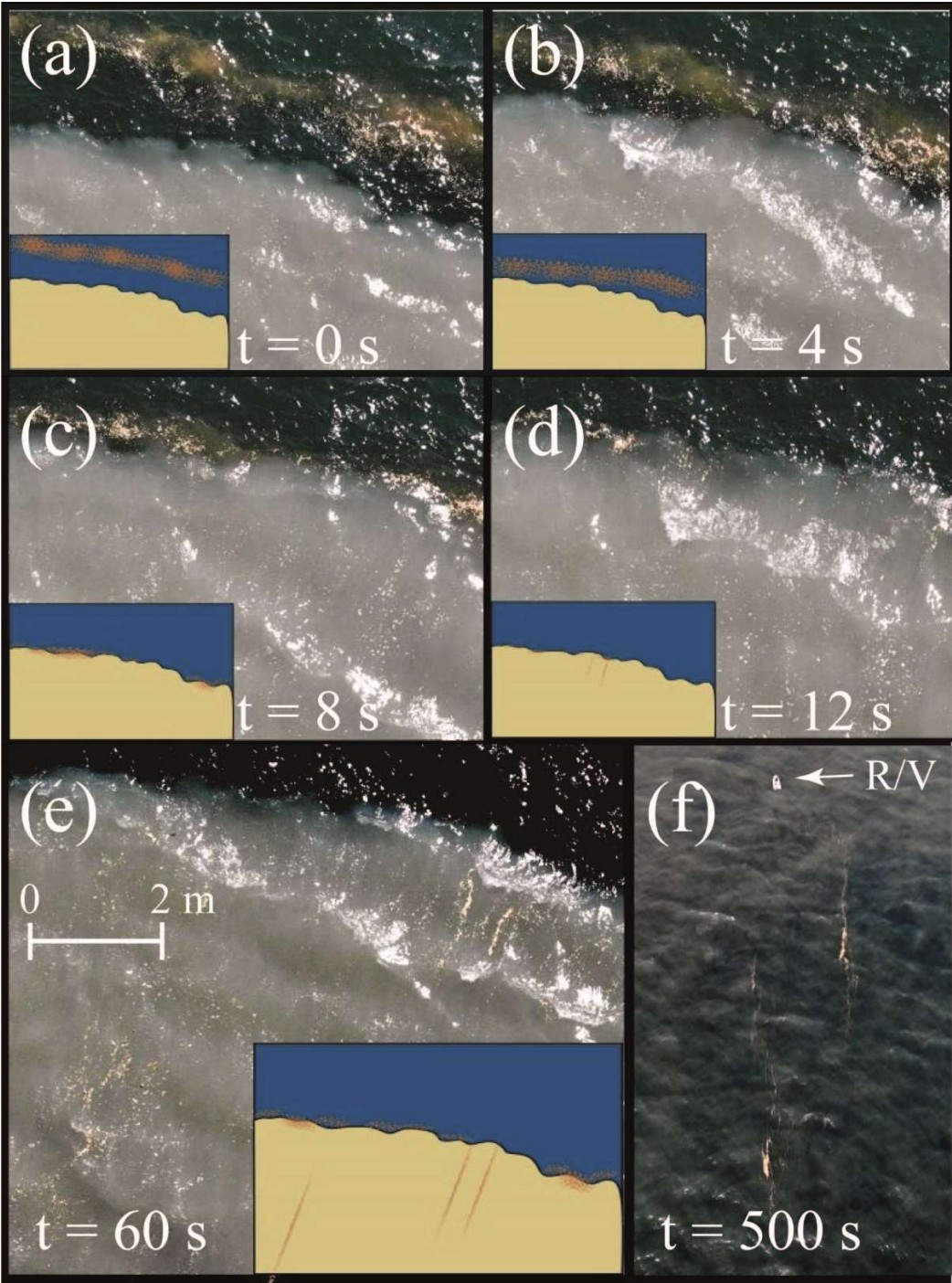

**Figure 10.** Aerial images of the Bzyb plume border on 15 April 2021 at (**a**) the beginning of the "cross-border" experiment, (**b**) after 4 s, (**c**) after 8 s, (**d**) after 12 s, (**e**) after 60 s, and (**f**) after 500 s illustrating the Stokes drift of sawdust particles across the plume-sea interface in the fine surface layer. The insets illustrate the location of sawdust particles (indicated by orange dots) in relation to the plume border. Note that panel (**f**) is not a nadir image and it has a significantly larger scale than those in panels (**a**–**d**).

## 4. Discussion

The aerial observations and in situ measurements described in the previous section provide new insights into the physical background of the lobe-and-cleft structures at the plume border, which are manifestations of frontal instabilities. Typically, the propagation rates of gravity currents are extremely fast and are considerably faster than the fluctuations

responsible for the mixing [39]. However, it is not the case with the relatively slow propagation of plume fronts, which provides the opportunity to examine the internal processes regulating instabilities at the front.

Based on our measurements, we could distinguish two types of frontal instabilities, which are formed at the plume border (Figure 11). The first type is generated in the near-field part of the plume in the presence of a large velocity shear at the plume-sea interface (>20–30 cm/s) (Figures 6a and 7). These instabilities are manifested by relatively small sizes of lobes (~3–7 m) and asymmetric vorticity (one gyre) within the lobes. This type of lobe-and-cleft structure is associated with the Kelvin–Helmholtz instabilities generated by velocity shear. The second type is generated at the far-field part of the plume without large velocity shear and has a large variety of sizes of lobes (~5–50 m) (Figures 6b, 8 and 9). These instabilities have symmetric vorticity patterns (two gyres) within the lobes and are associated with the Rayleigh–Taylor instabilities generated by a large pressure gradient across the plume-sea interface. The general scheme of formation of the Kelvin–Helmholtz and Rayleigh–Taylor instabilities at the plume border is shown in Figure 11.

According to the theory described in [46], the type of a frontal instability can be determined using the Richardson number $Ri = N^2/((\partial u/\partial z)^2 + (\partial v/\partial z)^2)$, where $N = \sqrt{db/dz}$ is the buoyancy frequency, $b = g(\rho_0 - \rho)/\rho$ is the buoyancy, $\rho_0$ and $\rho$ are the plume and sea densities, respectively, $u$ and $v$ are the $x$- and $y$-components of the horizontal velocity, respectively. The Kelvin–Helmholtz instabilities develop at the frontal zone when $0 < Ri < 0.25$, while the baroclinic instability dominates at $Ri > 0.95$ [46]. For the considered lateral borders of river plumes we have good agreement of the value of the Richardson number and the distinguished Kelvin–Helmholtz (with small size ~5–7 m and large velocity shear >20–30 cm/s) and Rayleigh–Taylor (with large size ~5–50 m and small velocity shear <10 cm/s) instabilities. Indeed, $Ri < 10^{-2}/4 \cdot 10^{-2} = 0.25$ for the Kelvin–Helmholtz instabilities and $Ri > 10^{-2}/10^{-2} = 1$ for the Rayleigh–Taylor instabilities.

In order to quantify the importance of the Coriolis force on the formation of instabilities, we calculated the Rossby number $Ro = U/L \cdot f$ for these instabilities, where $U$ is the velocity scale, $L$ is the horizontal length scale, $f$ is the Coriolis parameter. The resulting values are $Ro = 200$ for the Kelvin–Helmholtz instabilities and $Ro = 10$ for the Rayleigh–Taylor instabilities, which indicates the low significance of the Coriolis force for the considered processes. In addition, we calculated the Ertel potential vorticity for the frontal instabilities using the formula $q = q_v + q_h$, where $q_v = (f + \zeta) \cdot N^2$ is the baroclinic potential vorticity, $q_h = -\partial v/\partial z \times \partial b/\partial x + \partial u/\partial z \times \partial b/\partial y$ is the vertical potential vorticity, $\zeta = \partial v/\partial x - \partial u/\partial y$ is the vertical relative vorticity [49]. For the considered instabilities $f \sim 10^{-4}$, $\zeta \sim 10^{-2}–10^{-3}$, $N^2 \sim 10^{-2}$, therefore $q_v \sim 10^{-4}–10^{-5}$, $q_h \sim 10^{-5}$. As a result, $q_v > q_h$, which indicates that there are no symmetric instabilities at the plume lateral boundary [48–51].

The obtained results on the different physical backgrounds of near-field and far-field instabilities are in a good agreement with the recent study of Simpson et al. [41]. They also report the difference in sizes and vorticity patterns of lobe-and-cleft structures with and without velocity shear at the plume-sea interface, albeit that their numerical modeling limitedly addresses variability of sizes of these instabilities. Horner–Devine and Chickadel [33] speculate that the lobe-and-cleft structures at the plume border without velocity shear may result from the collapse of Kelvin–Helmholtz billows. In this case, the typical width of lobes is twice greater than the plume depth [47]. However, in this study we demonstrate that the width of lobes varies, by a wide range of ~5–50 m, and does not depend on the plume depth, which does not exceed 3–4 m at the Kodor and Bzyb plumes. This result supports the assumption that the lobe-and-cleft structures are generated by the Rayleigh–Taylor instabilities.

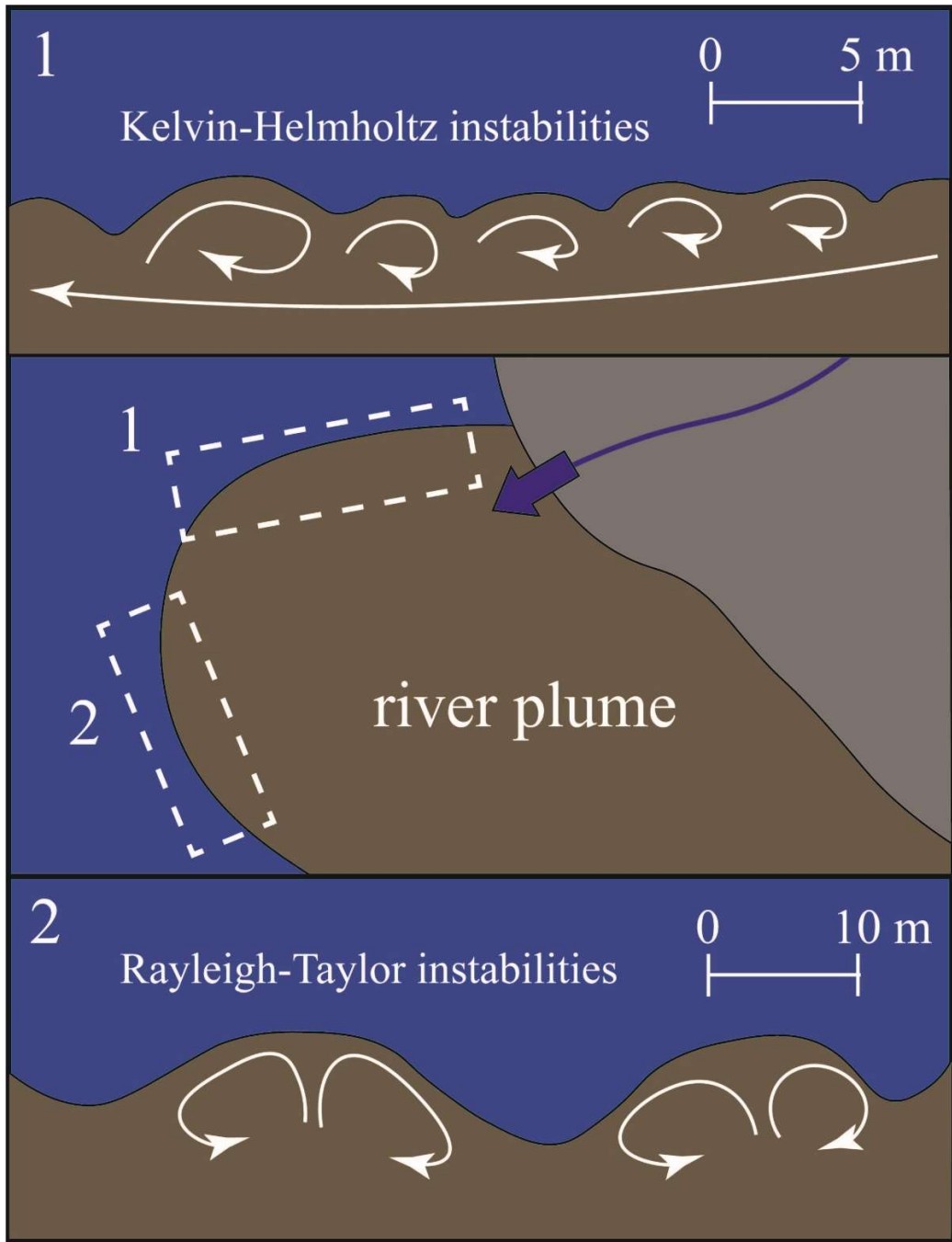

**Figure 11.** Scheme of formation and circulation of Kelvin–Helmholtz (**top**) and Rayleigh–Taylor (**bottom**) instabilities at different parts of a plume border (**middle**). The white dashed boxes and white numbers in the middle panel indicate location of zoomed areas of river plume border shown in the top (1) and bottom (2) panels.

In order to study the dependence of Rayleigh–Taylor instability on external forcing conditions, we used the Atwood number, which is prescribed as A = $(\rho_{sea} - \rho_{plume})/(\rho_{sea} + \rho_{plume})$, where $\rho_{sea}$ and $\rho_{plume}$ are the densities of the ambient sea and river plume, respectively. The Atwood number is widely used for non-dimensional parameterization of hydrodynamic instabilities in density stratified flows [52,53]. We calculated the Atwood number for all available in situ thermohaline measurements across the plume border, which were accompanied by aerial remote sensing of Rayleigh–Taylor instabilities during field surveys at the Kodor and Bzyb river plumes. The diameters of Rayleigh–Taylor instabilities for this dataset showed good linear relation with the values of the Atwood number (Figure 12). The

coefficient of determination $R^2$ for this relation is equal to 0.94 and the resulting formula for the diameter of Rayleigh–Taylor instabilities is $D = 66.6 - A \times 20.6$.

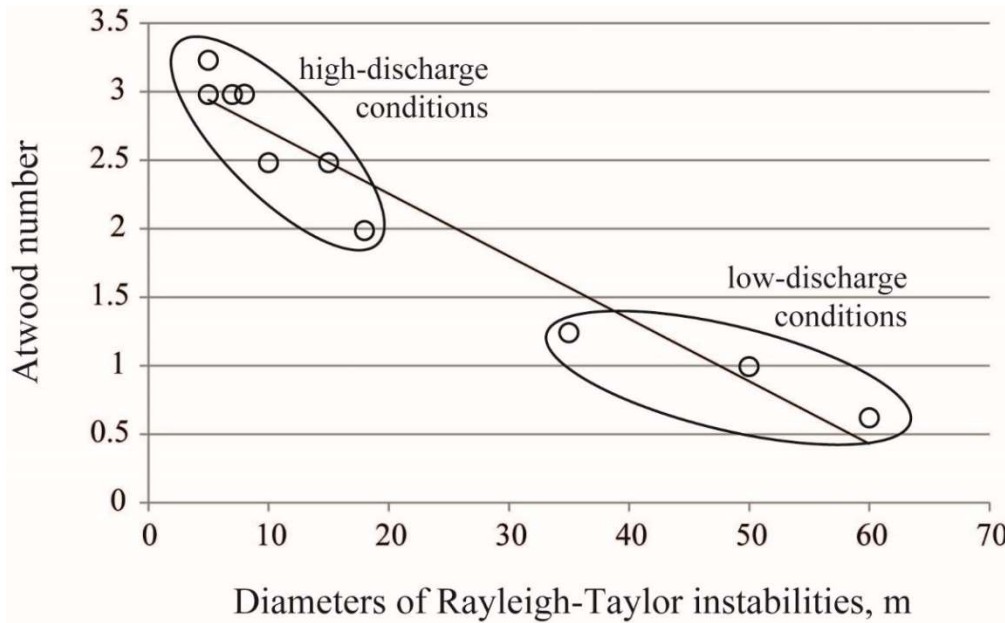

**Figure 12.** Dependence of diameters of Rayleigh–Taylor instabilities on the Atwood number. The black line represents linear regression.

The experiments with different types of tracers (corncobs and sawdust) at the plume-sea interface, as well as in situ thermohaline measurements provided qualitative assessments of influence of the Rayleigh–Taylor instabilities on transport of dissolved, suspended, and floating matter across sharp plume-sea interface. First, we observed distinct convergence at the plume borders, which resulted in sharp gradients of salinity and turbidity. This convergence was illustrated by quick advection of floating corncobs (~0.1–0.2 m/s) to the plume border and their aggregation at the borders. These corncobs remain at the plume border, albeit they could be further transported by circulation within the border instabilities. Second, in situ measurements and aerial observations reveal that the Rayleigh–Taylor instabilities are often accompanied by low-turbid and saline stripes within the plumes at small distance from plume border. We presume that merging of the Rayleigh–Taylor instabilities induce transport of suspended and dissolved matter across the plume borders, which results in formation of these low-turbid and saline stripes. Third, we registered the intense wind-driven transport of sawdust (~0.1 m/s) across the plume border, which was not hindered by border convergence. We explain this process by the Stokes drift, which effectively transports floating particles contained in the thin surface layer (several centimeters). Sawdust contained at a deeper sea layer was not transported across the plume border, on the contrary, it was advected below the plume as a result of border convergence. At the same time, the in situ measurements did not reveal any modification of salinity and turbidity structure of the plume-sea interface caused by the Stokes drift. Therefore, we presume that wind-driven Stokes drift limitedly affects transport of suspended and dissolved matter across the plume borders.

## 5. Conclusions

In this study, we describe aerial observations and synchronous in situ measurements focused on small-scale processes at the river plume–saline sea interface. For this purpose, we performed five field surveys at the small Kodor and Bzyp river plumes located at the eastern part of the Black Sea in 2018–2022. This study continues our previous research of small river plumes in the Black Sea focused on the structure of small plumes [2,3,57], influence of wind forcing and river discharge variability on small plumes [4,58,59], generation

of internal waves in river plumes [6,60], influence of small plumes on water quality and marine pollution [14,61].

The main findings of this study as follows. Manifestations of the Kelvin–Helmholtz and Rayleigh–Taylor instabilities at the plume borders are significantly different by size and vorticity patterns. The Kelvin–Helmholtz instabilities are small (~3–7 m) and have asymmetric vorticity (one gyre) within the lobes. These instabilities are formed as a result of velocity shear between river plume and ambient saline sea (>20–30 cm/s) and, therefore, are typical for near-field parts of small plumes. The Rayleigh–Taylor instabilities have large variety of sizes (5–50 m), symmetric vorticity (two gyres) and are formed by density gradient at the sharp plume border in case of low external forcing and low velocity shear. Typical sizes of the Rayleigh–Taylor instabilities linearly depend on the Atwood number, which represents the cross-border pressure gradient.

Formation and merging of the Rayleigh–Taylor instabilities induce intense transport of suspended and dissolved matter across the plume border. In the case of the Bzyb and Kodor plumes, this results in the formation of low-turbid and saline stripes within the plumes at a small distance from the plume border. Floating matter, on the contrary, is accumulated at the convergence zone at the plume border and is not affected by the instabilities. However, wind-driven Stokes drift could induce intense transport of floating matter across the plume border in a shallow (~2–3 cm) surface layer. This process could be an important issue for the spread of river-borne floating particles in the ocean as it could remove these particles from "dead ends" at convergence areas associated with frontal zones. At the same time, wind-driven Stokes drift has a limited effect on the transport of suspended and dissolved matter across the plume border and, therefore, does not modify the salinity and turbidity structure of the plume-sea interface.

The instabilities observed at the borders of small plumes have relatively small spatial scales. Moreover, frontal zones of small plumes are very thin (1–2 m) and narrow (<10 m) with sharp salinity gradients. Both these conditions require very small horizontal and vertical grid spacing of numerical models. As a result, general ocean models do not represent these processes due to insufficient spatial resolution. This fact highlights the importance of in situ studies of small plume instabilities, which is the case of this work. Borders of large river plumes have significantly greater horizontal and vertical extents, which could be represented by ocean reanalysis. However, the instabilities, which are formed at the small plume borders, are not observed at the borders of large plumes, which have much lower salinity and velocity gradients.

In this study, we focused mainly on the primary forcing, which generates plume border instabilities (pressure gradient and velocity shear). Secondary effects, including coastal currents, waves, and wind may have very strong effects on the plume boundary and the instabilities. However, these processes remained mostly unaddressed in this work due to the following reasons. Numerical modeling studies provide the opportunity to distinguish effectively the primary forcing and the secondary effects. However, it is much more complicated to perform similar studies using in situ measurements and observations. The presence of strong secondary effects (the intensity of which cannot be controlled as it can in numerical modeling studies) can completely blur and distort the primary forcing. Moreover, it is very complicated to distinguish the influence of different secondary forcing conditions, e.g., wind and waves, which are often present at the same time.

In order to obtain the baseline for understanding the general aspects of plume border instabilities we processed and analyzed experiments during low wind, wave, and coastal current forcing. During our field surveys, we also performed in situ measurements and aerial observations under significant wind, wave, and coastal current forcing conditions. In this paper, we describe and analyze only one of these experiments, i.e., the impact of wind-induced Stokes drift on the cross-border transport in the surface layer. Analysis of other experiments, as well as organization of additional process-oriented measurements and observations, is within the scope of future work.

**Author Contributions:** Conceptualization, A.O.; methodology, A.O. and A.G.; measurements, A.O., A.G., A.B., R.S., V.R. and R.Z.; formal analysis, A.O. and A.G.; investigation, A.O., A.G., A.B., R.S., V.R., R.Z. and R.D.; writing, A.O., A.G., A.B., R.S., V.R., R.Z. and R.D. All authors have read and agreed to the published version of the manuscript.

**Funding:** This research was funded by the Ministry of Science and Higher Education of the Russian Federation, theme FMWE-2021-0001 (collecting of in situ data), the Russian Science Foundation, research project 18-17-00156 (study of river plumes), and the Russian Foundation for Basic Research, research project 19-55-80004 (collecting and processing of aerial remote sensing data).

**Data Availability Statement:** The in situ data and aerial imagery are available at https://doi.org/10.5281/zenodo.6656838 (accessed on 22 June 2022).

**Conflicts of Interest:** The authors declare no conflict of interest.

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
