# Peer review of "Lateral Border of a Small River Plume: Salinity Structure, Instabilities and Mass Transport"

_remotesensing, doi:10.3390/rs14153818_

Round 1

Reviewer 1 Report

The manuscript is devoted to field measurements at the plume-sea boundaries using quadcopters equipped with an optical camera. The experiments were carried out at plume boundaries of two small rivers Kodor and Bzyp flowing into the Black Sea. The Authors already have an article published in Remote Sensing in 2020 on the same topic. The present manuscript discusses the same experiments plus experiments carried out in 2021-2022.

The use of quadcopter optical data makes it possible to study processes that are difficult to study from space and cannot be studied with in-situ measurements. However, considering the large volume of experimental data collected, I would expect more profound scientific conclusions rather than a set of nice pictures.

My comments are as follows:

1.           The reasoning regarding the type of instability is not convincing, especially the Kelvin-Helmholtz instability. Moreover, nothing is said on the effect that coastal current, waves, and wind may have on the plume boundary and the instabilities they may induce. It is only mentioned that waves and wind were weak. What about the impact of the bottom relief?

2.           From Data and Methods it follows that an ADCP was used. However, no ADCP measurements are presented in Results while they would be useful when discussing the effect of coastal current on the formation of instability at the plume boundary.

3.           The title of the article lists Salinity Structure in the first place among the studied processes. Nevertheless only Fig. 4 with a short commentary is devoted to salinity structure in the manuscript. It would be of great interest to learn what new information on the matter the measurements have brought about.

4.           Lines 154-159 give the values of discharge. It is a very important parameter on which the size of the plume mostly depends (beside wind forcing, currents, etc.) Where do these values come from? Were the data from coastal gauges used, which ones? Or are these the Authors’ own measurements, which ones? This information should be stated in Data and Methods.

5.           Of 51 works on the list of references, 18 belong to the Authors (every 3rd). And not all of them are directly related to the topic of the manuscript. This looks like deliberately promoting their work and boosting  citations. Only references directly related to the discussed topic should be left on the list.

6.           Fig. 9 caption does not match the figure notation:  Kelvin-Helmholtz and Reyleigh-Taylor are mixed up.

Author Response

The manuscript is devoted to field measurements at the plume-sea boundaries using quadcopters equipped with an optical camera. The experiments were carried out at plume boundaries of two small rivers Kodor and Bzyp flowing into the Black Sea. The Authors already have an article published in Remote Sensing in 2020 on the same topic. The present manuscript discusses the same experiments plus experiments carried out in 2021-2022.

The use of quadcopter optical data makes it possible to study processes that are difficult to study from space and cannot be studied with in-situ measurements. However, considering the large volume of experimental data collected, I would expect more profound scientific conclusions rather than a set of nice pictures.

My comments are as follows:

  1. The reasoning regarding the type of instability is not convincing, especially the Kelvin-Helmholtz instability.

Many thanks for this comment. According to theory described in (Stone, 1966), the type of frontal instability can be determined using the Richardson number Ri = N2 / ((∂u/∂z)2 + (∂v/∂z)2), where N =  is the buoyancy frequency, b = g(ρ0 – ρ) / ρ is the buoyancy, u and v are the x- and y-components of horizontal velocity. Kelvin-Helmholtz instabilities develop at the frontal zone when 0 < Ri < 0.25, while baroclinic instability dominates at Ri > 0.95. For the considered lateral borders of river plumes we have good agreement of the value of the Richardson number and the distinguished Kelvin–Helmholtz (with small size ~5-7 m and large velocity shear >20-30 cm/s) and Rayleigh-Taylor (with large size ~5-50 m and small velocity shear <10 cm/s) instabilities. Indeed, the Ri < 10-2 / 4·10-2 = 0.25 for the Kelvin–Helmholtz instabilities and Ri > 10-2 / 10-2 = 1 for the Rayleigh-Taylor instabilities. This clarification was added to the text.

Stone, P., 1966. On non-geostrophic baroclinic stability. J. Atmos. Sci., 23, 390—400. doi: 10.1175/1520-0469(1966)023<0390:ONGBS>2.0.CO;2

Moreover, nothing is said on the effect that coastal current, waves, and wind may have on the plume boundary and the instabilities they may induce. It is only mentioned that waves and wind were weak. What about the impact of the bottom relief?

We totally agree that coastal current, waves, and wind may have very strong effect on the plume boundary and the instabilities. However, in this work, we do not focus on these forcing conditions due to the following reasons. Numerical modeling studies provide opportunity to effectively distinguish the primary forcing, that generates plume border instabilities (i.e., pressure gradient and velocity shear), and the secondary effects of coastal current, waves, and wind. However, it is much more complicated to perform similar studies using in situ measurements and observations. Presence of strong secondary effects (which intensity can not be controlled as in numerical modeling studies) can completely blur and distort the primary forcing. Moreover, it is very complicated to distinguish the influence of different secondary forcing conditions, e.g., wind and waves, which are often present at the same time. Therefore, in this study, we focused only on primary forcing to provide the baseline for understanding the general aspects of plume border instabilities. For this reason, we processed and analyzed only experiments during low wind, wave, and coastal current forcing.

We want to add, that during our field surveys we also performed in situ measurements and aerial observations under significant wind, wave, and coastal current forcing conditions. In this paper, we describe and analyze only one of these experiments, i.e., the impact of wind-induced Stokes drift on cross-border transport in the surface layer. This result was unexpected, albeit we could explain it because the Stokes drift did not distort the formation of plume border instabilities. Other experiments provided more puzzling and hard-to-explain results. Analysis of these experiments, as well as organization of additional process-oriented measurements and observations is within the scope of future work. The related clarification was added to the text.

  1. From Data and Methods it follows that an ADCP was used. However, no ADCP measurements are presented in Results while they would be useful when discussing the effect of coastal current on the formation of instability at the plume boundary.
  2. The title of the article lists Salinity Structure in the first place among the studied processes. Nevertheless only Fig. 4 with a short commentary is devoted to salinity structure in the manuscript. It would be of great interest to learn what new information on the matter the measurements have brought about.

Many thanks for these comments. According to your recommendations, we added the figures and text describing more CTD (Figures 4 and 5) and ADCP (Figure 6) data collected across the lateral plume border. We added the text describing the salinity and velocity structure at the near-field and far-field parts of the plume with large and small velocity shear to support our study of frontal instabilities. Also, we want to state the we are currently preparing a new paper specifically focused on the salinity and velocity structure at the small and large river plumes. In particular, in this new paper we will describe measurements at the Bzyb and Kodor plumes, which were not covered in the current paper.

  1. Lines 154-159 give the values of discharge. It is a very important parameter on which the size of the plume mostly depends (beside wind forcing, currents, etc.) Where do these values come from? Were the data from coastal gauges used, which ones? Or are these the Authors’ own measurements, which ones? This information should be stated in Data and Methods.

Many thanks for this comment, the methodology of river discharge measurement indeed was not described in the manuscript. The discharge values were reconstructed using the indirect method based on satellite observations and Lagrangian numerical modeling of river plumes. The general idea of this method is reconstruction of configuration of external forcing conditions (including river discharge rate) for a numerical model, which provides a river plume similar to that observed at satellite imagery. This method was validated against gauge measurements at small rivers in the study area, which was described in detail in (Osadchiev A.A. A method for quantifying freshwater discharge rates from satellite observations and Lagrangian numerical modeling of river plumes. Environmental Research Letters. 2015, 10, 085009. doi:10.1088/1748-9326/10/8/085009). The method was additionally validated for the Kodor plume by in situ discharge measurements performed during a hydrological field survey in August – September 2018. The related clarification was added to the Data and Methods section.

  1. Of 51 works on the list of references, 18 belong to the Authors (every 3rd). And not all of them are directly related to the topic of the manuscript. This looks like deliberately promoting their work and boosting citations. Only references directly related to the discussed topic should be left on the list.

We agree that the number of self-citations is too high in this paper, so we reduced their number from 18 to 10. The remaining citations have direct relation to the topic and the study area.

  1. Fig. 9 caption does not match the figure notation: Kelvin-Helmholtz and Reyleigh-Taylor are mixed up.

Corrected.

Reviewer 2 Report

Overview:

The authors use high-resolution aerial remote sensing supported by in situ measurements to study small scale instabilities. They describe their spatial and temporal characteristics and then reconstruct their relation to density gradient and velocity shear. The manuscript is well written and the figures are neat. The results in this paper would contribute to future study of regional circulation dynamics in this region. I recommend minor revision.

Minor revisions:

1)    Could you estimate the Rossby number of these instabilities?

2)    How about calculate the Ertiel potential vortiticy (EPV) to evalute the frontal instabilities?

3)     Does any reanalysis ocean model resolve these instabilities in the river plume?

Author Response

Overview:

The authors use high-resolution aerial remote sensing supported by in situ measurements to study small scale instabilities. They describe their spatial and temporal characteristics and then reconstruct their relation to density gradient and velocity shear. The manuscript is well written and the figures are neat. The results in this paper would contribute to future study of regional circulation dynamics in this region. I recommend minor revision.

 Minor revisions:

1) Could you estimate the Rossby number of these instabilities?

According to your recommendation, we calculated the Rossby number Ro = U / L·f for these instabilities, where U is the velocity scale (equal to ~0.1 m/s for Kelvin–Helmholtz instabilities and ~0.01 m/s for Rayleigh-Taylor instabilities), L is the horizontal length scale (equal to ~5 m for Kelvin–Helmholtz instabilities and ~10 m for Rayleigh-Taylor instabilities), f is the Coriolis parameter (equal to 0.992·10-4 1/s in the study area). The resulting values are Ro = 200 for the Kelvin–Helmholtz instabilities and Ro = 10 for the Rayleigh-Taylor instabilities.

2) How about calculate the Ertiel potential vortiticy (EPV) to evalute the frontal instabilities?

Also, according to your recommendation, we calculated the Ertel potential vorticity for the frontal instabilities using the formula q = qv + qh, where qv = (f + ζN2 is the baroclinic potential vorticity, qh = – ∂v/∂z · ∂b/∂x + ∂u/∂z · ∂b/∂y is the vertical potential vorticity, ζ = ∂v/∂x∂u/∂y is the vertical relative vorticity, b = g(ρ0 – ρ) / ρ is the buoyancy, N =  is the buoyancy frequency, u and v are the x- and y-components of horizontal velocity. For the considered instabilities f ~ 10-4, ζ ~ 10-2 – 10-3, N2 ~ 10-2, qv ~ 10-4 – 10-5, qh ~ 10-5. As a result, qv > qh, which indicates that there are no symmetric instabilities at the plume lateral boundary.

3) Does any reanalysis ocean model resolve these instabilities in the river plume?

The instabilities observed at the small plume borders have relatively small spatial scales, namely, 5-50 m. Moreover, frontal zones of small plumes are very thin (1-2 m) and narrow (<10 m) with sharp salinity gradients. Both these conditions require very small horizontal and vertical grid spacing of numerical models. To the extent of our knowledge, the general reanalysis ocean models have horizontal spatial resolution greater than several km, which is not enough to represent the instabilities at the small plume borders. Borders of large river plumes have significantly greater horizontal and vertical extents, which could be represented by reanalysis ocean model. However, the instabilities, which are formed at the small plume borders and are described in our study, are not observed at the borders of large plumes, which have much lower salinity and velocity gradients. This clarification was added to the text.

Reviewer 3 Report

1、In Introduction, please describe the reasons for there is a large gap between theoretical solutions, numerical and laboratory modeling of frontal instabilities at stratified fluids, on the one hand, and observations and measurements of frontal instabilities in natural systems in detail.

2、The author focused on several indicators of density gradient and velocity shear. please demonstrate whether the indicators are sufficient, other scholars used these indicators in their research, and what are their results.

3、Why did the author choose candy edge detection? There are many excellent machine learning algorithms for water boundary extraction, such as Markov Random Field model, Active Contour Model, Random Forests and so on.

4、The paper lacks the necessary flow chart to introduce the key steps of the whole experimental process, which makes it more intuitive for readers to understand the logic of the whole scientific research work in this paper.

Author Response

1. In Introduction, please describe the reasons for ‘there is a large gap between theoretical solutions, numerical and laboratory modeling of frontal instabilities at stratified fluids, on the one hand, and observations and measurements of frontal instabilities in natural systems’ in detail.

We agree that this statement should be regarded only to river plumes, not to natural systems in general. Also, we provide the following explanation for this statement. In situ measurements represent ground truth for studies of processes in river plumes. On the other hand, spatial and/or temporal resolution of discrete in situ measurements is relatively low. As a result, studies based only on in situ measurements often have the inherent spatial or temporal limitations. Satellite and aerial remote sensing could support in situ measurements and substantially increase spatial coverage of the considered processes including frontal instabilities. However, remote sensing observations are limited to surface manifestations of these processes and do not resolve the vertical structure of river plumes. On the opposite, theoretical solutions, numerical and laboratory modeling reproduces the three-dimensional plume structure with relatively high spatial and temporal resolution. The main limitation of these studies consists in fact that they require thorough validation against in situ data (which is often lacking) in order to verify that they represent processes which occur in real river plumes.

2. The author focused on several indicators of density gradient and velocity shear. please demonstrate whether the indicators are sufficient, other scholars used these indicators in their research, and what are their results.

Many thanks for this point. The main straightforward characteristics, which are used to study instabilities in river plumes, include, first, the spatial and temporal characteristics of these instabilities (wavelength, amplitude, motion speed, vorticity, residual time, etc.) and, second, the characteristics of gradients at the plume-sea interface, which govern formation of instabilities (density gradient and velocity shear) (Trump and Marmorino, 2003; Horner-Devine and Chickadel, 2017; Simpson et al., 2022). A large set of characteristics, which describe the role of instabilities in turbulent mixing (eddy viscosity, vertical diffusivity, eddy kinetic energy, buoyancy dissipation, etc.), is mainly addressed and analyzed in numerical modeling studies due to complexity of in situ turbulence measurements in river plumes (Tedford et al., 2009; Iwanaka et al., 2018; Auoche et al., 2020; 2021; 2022). Finally, a number of dimensionless numbers (Rossby, Richardson, Reynolds, Atwood, Grashof, Ertel, etc.) are widely used to compare mass, momentum, and energy transport associated with instabilities in different laboratory and natural systems (Stone, 1966; Thomas et al., 2013; White and Helfrich, 2013; Holmes et al., 2014; Xie et al., 2019; Dong et al., 2021; Yu et al., 2021; Dai and Huang, 2022). The related clarification was added to the text.

3. Why did the author choose candy edge detection? There are many excellent machine learning algorithms for water boundary extraction, such as Markov Random Field model, Active Contour Model, Random Forests and so on.

The Canny edge detection algorithm was chosen due to two main reasons: it has low computational cost and it demonstrated accurate detection for our dataset. The dataset consisted of pictures, which generally contain only two “objects”: turbid (brown) plume and clean (blue) seawater. These “objects” have large difference in color, so the typical picture was a two-colored image with clear simple-shaped border. As the brightness gradient between these two water masses was high, the border was efficiently detected by the candy edge algorithm. What is more important, the Canny edge detector was relatively insensible to local minima, which was frequently registered at the plume-sea border due to increased turbulent mixing and presence of floating matter and foam. The related clarification was added to the text.

4. The paper lacks the necessary flow chart to introduce the key steps of the whole experimental process, which makes it more intuitive for readers to understand the logic of the whole scientific research work in this paper.

Many thanks for this point. We added the related information to Data and Methods section. The general work flow chart of our field surveys is the following. First, we performed aerial remote sensing of the plume using quadcopters. Aerial observations provided the initial information about the plume shape, location of its borders, and presence of border instabilities. Then we collected aerial imagery and/or video records of detected instabilities. Second, synchronously to remote sensing, we performed in situ measurements of thermohaline and velocity structure across the plume border. CTD measurement were organized either as surface to bottom profiling along the cross-border transect with high spatial resolution or continuous measurements at fixed depths during drift of a boat across the plume border. ADCP measurements were performed either from stable bottom-mounted profiler or moving boat-mounted profiler with bottom tracking. Third, we scattered floating natural tracers (sawdust and corncobs) from a boat at different parts of the plume-sea interface and were tracing their motion by aerial remote sensing. The resulting data set collected during field surveys provided the necessary information about thermohaline and velocity structure across the plume border, as well as spatial scales, circulation patterns, and velocities of plume border instabilities.

Round 2

Reviewer 1 Report

The Authors have thoroughly revised and improved the manuscript and provided satisfactory explanations.